# TNF/IFN-γ Co-Signaling Induces Differential Cellular Activation in COVID-19 Patients: Implications for Patient Outcomes

**DOI:** 10.3390/ijms26031139

**Published:** 2025-01-28

**Authors:** Lucero A. Ramón-Luing, Laura Edith Martínez-Gómez, Carlos Martinez-Armenta, Gabriela Angélica Martínez-Nava, Karen Medina-Quero, Gloria Pérez-Rubio, Ramcés Falfán-Valencia, Ivette Buendia-Roldan, Julio Flores-Gonzalez, Ranferi Ocaña-Guzmán, Moisés Selman, Alberto López-Reyes, Leslie Chavez-Galan

**Affiliations:** 1Instituto Nacional de Enfermedades Respiratorias Ismael Cosío Villegas, Mexico City 14080, Mexico; ramonluing@yahoo.com.mx (L.A.R.-L.); glofos@yahoo.com.mx (G.P.-R.); dcb_rfalfanv@hotmail.com (R.F.-V.); ivettebu@yahoo.com.mx (I.B.-R.); juliofglez@gmail.com (J.F.-G.); arocana@iner.gob.mx (R.O.-G.); mselmanl@yahoo.com.mx (M.S.); 2Laboratorio de Gerociencias, Instituto Nacional de Rehabilitación Luis Guillermo Ibarra Ibarra, Mexico City 14080, Mexico; laurae.mtzg@gmail.com (L.E.M.-G.); c.armenta1208@gmail.com (C.M.-A.); ameria.justice@gmail.com (G.A.M.-N.); allorey@yahoo.com (A.L.-R.); 3Immunology Laboratory, Escuela Militar de Graduados de Sanidad, Mexico City 11200, Mexico; kmq.kmq5@gmail.com

**Keywords:** TNF, IFN-γ, COVID-19, inflammation, cell death, intracellular TLR

## Abstract

TNF and IFN-γ are key proinflammatory cytokines implicated in the pathophysiology of COVID-19. Toll-like receptor (TLR)7 and TLR8 are known to recognize SARS-CoV-2 and induce TNF and IFN-γ production. However, it is unclear whether TNF and IFN-γ levels are altered through TLR-dependent pathways and whether these pathways mediate disease severity during COVID-19. This study aimed to investigate the association between TNF/IFN-γ levels and immune cell activation to understand their role in disease severity better. We enrolled 150 COVID-19 patients, who were classified by their systemic TNF and IFN-γ levels (high (H) or normal–low (N-L)) as TNF^H^IFNγ^H^, TNF^H^IFNγ^N-L^, TNF^N-L^IFNγ^H^, and TNF^N-L^IFNγ^N-L^. Compared to patients with TNF^N-L^IFNγ^N-L^, patients with TNF^H^IFNγ^H^ had high systemic levels of pro- and anti-inflammatory cytokines and cytotoxic molecules, and their T cells and monocytes expressed TNF receptor 1 (TNFR1). Patients with TNF^H^IFNγ^H^ presented the SNP rs3853839 to TLR7 and increased levels of MYD88, NFκB, and IRF7 (TLR signaling), FADD, and TRADD (TNFR1 signaling). Moreover, critical patients were observed in the four COVID-19 groups, but patients with TNF^H^IFNγ^H^ or TNF^H^IFNγ^N-L^ most required invasive mechanical ventilation. We concluded that increased TNF/IFN-γ levels are associated with hyperactive immune cells, whereas normal/low levels are associated with hypoactivity, suggesting a model to explain that the pathophysiology of critical COVID-19 may be mediated through different pathways depending on TNF and IFN-γ levels. These findings highlight the potential for exploring the modulation of TNF and IFN-γ as a therapeutic strategy in severe COVID-19.

## 1. Introduction

Toll-like receptors (TLRs) are one of the families of pattern recognition receptors expressed in immune cells that recognize pathogen-associated molecular patterns. Eleven types of TLRs have been found in human cells, and they play a central role in mediating the innate immune response through the delivery of proinflammatory cytokines, such as tumor necrosis factor (TNF) and interferon-gamma (IFN-γ) [1,2,3].

TLR7 and TLR8 are exclusively intracellular and expressed on the endosome membrane; both act as sensors of viral single-stranded ribonucleic acid (ssRNA) and have been implicated in antiviral defense, triggering the synthesis of proinflammatory cytokines through the myeloid differentiation primary response gene 88 (MyD88)-dependent signaling pathway [4,5]. Furthermore, through TLR8, TNF-receptor-associated factor (TRAF)-6 is recruited to activate the nuclear factor kappa light-chain enhancer of activated B cells (NFκB) for inducing inflammatory cytokines, i.e., interleukin (IL)6, IL1β, TNF, and IFN-γ; TLR7 and TLR8 use TRAF3 to activate interferon regulatory factors (IRFs) 7 and 3 and induce type I IFN synthesis (IFN-α and IFN-β) [6,7].

Also, TLR signaling pathways have been implicated as key regulators of the pathogenesis of severe acute respiratory syndrome coronavirus 2 (SARS-CoV-2), a positive-sense ssRNA betacoronavirus causing coronavirus disease 2019 (COVID-19) [3,4,8]. A bioinformatic analysis revealed ssRNA fragments in the SARS-CoV-2 genome that could be recognized by TLR7 and TLR8, triggering innate immune hyperactivation [9]. As TLR expression is determined by genetic variation within the TLR genes [10], the study of single-nucleotide polymorphisms (SNPs) is essential to understand how host genetic variation plays a role in COVID-19 infection, clinical presentations, and the host immune response.

In COVID-19, the induction of a cytokine storm is mainly regulated by TLR3, TLR4, TLR7, and TLR8 and triggers an excessive inflammatory response, leading to severe disease and death [11]. TLRs triggered by SARS-CoV-2 lead to inflammasome activation and IL-1β secretion, which induces IL-6 production [12]. The TNF/TNF receptor 1 (TNFR1) axis has also been associated with mortality; high TNF receptor (TNFR1 and TNFR2) soluble levels are found in COVID-19 patients, although only TNFR1 is related to severity and mortality [13]. TNF/TNFR1 signaling can induce cell death or an inflammatory response and survival [14]. Recently, we reported that patients with severe COVID-19 displaying high TNF and IFN-γ levels show a hallmark of PANoptosis associated with an exacerbated proinflammatory profile [15].

To the best of our knowledge, the influence of TLRs on TNF and IFN-γ production and the implications for patient outcomes remain unclear. This study aimed to identify whether TNF and IFN-γ levels play a central role in regulating the activated state of immune cells in COVID-19 patients and explore potential links with TLR7 and TLR8 signaling.

## 2. Results

### 2.1. Baseline Characteristics of COVID-19 Patients Classified into Four Groups Based on TNF and IFN-γ Levels

Table 1 summarizes the baseline characteristics of 125 COVID-19 patients categorized into four groups based on TNF and IFN-γ levels. The median age of all patients was 56 years (IQR 45–64). The COVID-19 groups exhibited similar baseline characteristics, except for D-dimer, which was higher in the TNF^H^IFNγ^N-L^ group (mean 3.1 ng/mL) compared to the TNF^N-L^IFNγ^H^ (*p* < 0.01) and TNF^N-L^IFNγ^N-L^ (*p* < 0.0556) groups.

### 2.2. The TNF^H^IFNγ^H^ Group Has Higher Levels of Proinflammatory and Anti-Inflammatory Cytokines and Cytotoxic Molecules Than the TNF^N-L^IFNγ^N-L^ Group

Regarding proinflammatory cytokines, IL-2 and IL-6 levels were higher in the TNF^H^IFNγ^H^ group than in the TNF^N-L^IFNγ^N-L^ group (*p* < 0.01 for IL-2 and *p* < 0.05 for IL-6) (Figure 1A). Regarding anti-inflammatory cytokines, IL-10 levels were higher in the TNF^H^IFNγ^H^ group than in the TNF^N-L^IFNγ^N-L^ group (*p* < 0.05) (Figure 1B). Despite differences, the four COVID-19 groups had higher cytokine levels than the HD group (dotted red line).

Regarding cytotoxic molecules, sFas, sFasL, granzyme A, granzyme B, perforin, and granulysin levels were significantly higher in the TNF^H^IFNγ^H^ group than in the TNF^N-L^IFNγ^N-L^ group (*p* < 0.05 for sFas, sFasL, granzyme A, granzyme B, and perforin; *p* < 0.001 for granulysin) (Figure 1C). Except for perforin, COVID-19 patients produced high levels of cytotoxic molecules compared to the HD group (dotted red line).

### 2.3. IMV Patients Exhibit a High Frequency of T Cell and Monocyte TNFR+1 and Hyperactive Production of Cytokines and Cytotoxic Molecules

Among TNF^H^IFNγ^H^ patients, 59% were classified as critical, and approximately 80% of patients with high TNF levels (TNF^H^IFNγ^H^ and TNF^H^IFNγ^N-L^) required invasive mechanical ventilation (IMV). Conversely, 80% of patients in the TNF^N-L^IFNγ^H^ and TNF^N-L^IFNγ^N-L^ groups did not require invasive mechanical ventilation (NIMV) (Table 1).

An additional patient cohort was divided into IMV and NIMV groups (Appendix A) to evaluate the TNF/TNFR axis and hyperactivity. Data revealed that the frequency of CD4+TNFR1+ T cells was significantly higher in IMV patients compared to NIMV patients (*p* < 0.01) or the HD group (*p* < 0.0001). Similarly, the frequency of CD8+TNFR1+ T cells was elevated in IMV patients compared to NIMV patients (*p* < 0.05) or the HD group (*p* < 0.01). TNFR2 expression on T cells remained unchanged (Figure 2A).

As observed in T cells, the frequency of CD14+TNFR1+ monocytes was higher in IMV patients than in NIMV patients (*p* < 0.05) or the HD group (*p* < 0.01). Notably, TNFR2 expression on CD14+ monocytes was also elevated in IMV patients compared to the HD group (*p* < 0.0001) (Figure 2B).

We confirmed that IMV patients showed a similar profile of soluble proteins to patients with TNF^H^IFNγ^H^ and NIMV patients than patients with TNF^H^IFNγ^N-L^; data showed that IMV patients had higher levels of IL-6 (*p* < 0.001) and IL-10 (*p* < 0.05) than NIMV patients. Moreover, compared to NIMV patients, IMV patients had higher levels of sFas (*p* < 0.01), perforin (*p* < 0.05), and granulysin (*p* < 0.01) (Figure 2C). 

### 2.4. SNPs in TLR7 and TLR8 Genes Are Related to TNF and IFN-γ Levels in COVID-19 Patients

Appendix A shows the allelic frequencies of 122 individuals. We observed significant differences between the alleles of the SNP rs3853839 in *TLR7*, with the C allele being more frequent in patients with TNF^H^IFNγ^H^.

Similar allele frequencies were found in the SNPs of *TRL8*; thus, linkage disequilibrium was performed. We found an r-square value of 0.85, performed haplotypes GGG and ACA, and indicated D’ pairwise in the diamonds (Figure 3A). Only patients with TNF^H^IFNγ^N-L^ showed a dominance of haplotype ACA; the other groups had a predominance of haplotype GGG (Appendix A). SNPs in the *ACE2* gene did not show a dominant frequency in any COVID-19 group.

### 2.5. SNPs in TLR7 and TLR8 Could Play a Role in Increasing TNF and IFN-γ Levels in COVID-19 Patients

Evaluation of the association of SNPs with groups displaying high cytokine levels showed that the rs2285666 (*ACE2* gene) allele T is present mainly in patients with higher TNF levels (median 9.0 pg/mL, IQR 3.9–15.7) than in those with the C allele (*p* < 0.05) (Figure 3B). Regarding *TLR7* polymorphisms, the rs179008 T allele and the rs3853839 G allele were present in patients with higher IFN-γ levels, specifically to rs179008 T *p* < 0.05 (median 28 pg/mL, IQR 11.4–57.9), and rs3853839 G *p* < 0.01 (median 30 pg/mL, IQR 3.5–57.9) (Figure 3C,D).

The haplotype ACA of *TLR8* was also present in patients with higher IFN-γ levels (median 31 pg/mL, IQR 3.5–57.9) (*p* < 0.01) (Figure 3E).

A sex-stratified analysis of soluble levels of IFN-γ and TNF revealed that in men (n = 84), there were significant differences in TNF concentrations in rs2285666 (*ACE2* gene) and the ACA haplotype. Significant differences between rs3853839 (*TLR7*) and the ACA haplotype were observed in the IFN-γ concentration. However, after adjusting with the Bonferroni test, we observed statistical significance in rs2285666 (*ACE2* gene) and rs3853839 (*TLR7* gene). In women (n = 38), we observed significant differences between rs179009 and the ACA haplotype. There was no statistical significance with the adjustment of *p* by a multiple-comparison Bonferroni test (Appendix A).

### 2.6. TNF^H^IFNγ^H^ Patients Have Higher Levels than TNF^N-L^IFNγ^N-L^ Patients of Molecules Associated with TLR Signaling and Cell Death Mediated by the TNF/TNFR1 Pathway

Compared to the TNF^H^IFNγ^H^ group, the TNF^N-L^IFNγ^N-L^ group showed lower levels of *MYD88* (*p* < 0.001), *NFKB1* (*p* < 0.001), and *IRF7* (*p* < 0.01). The TNF^N-L^IFNγ^H^ group also showed higher levels than the TNF^N-L^IFNγ^N-L^ group of *MYD88* (*p* < 0.0001), *NFκB1* (*p* < 0.001), and *IRF7* (*p* < 0.001) (Figure 4A–C). Patients with only increased TNF did not show differences from the other groups.

Regarding molecules involved in TNF/TNFR1 signaling, the TNF^N-L^IFNγ^H^ group had increased *TRAF2* levels compared to TNF^H^IFNγ^H^ (*p* < 0.05) and TNF^H^IFNγ^N-L^ (*p* < 0.001) groups. The TNF^N-L^IFNγ^N-L^ group showed lower levels of *FADD* than TNF^H^IFNγ^H^ (*p* < 0.05), TNF^H^IFNγ^N-L^ (*p* < 0.01), and TNF^N-L^IFNγ^H^ (*p* < 0.001) groups. The *TRADD* level was lower in the TNF^N-L^IFNγ^N-L^ group than in TNF^H^IFNγ^H^ (*p* < 0.01) and TNF^N-L^IFNγ^H^ (*p* < 0.001) groups; the TNF^N-L^IFNγ^H^ group also showed lower *TRADD* levels than the TNF^N-L^IFNγ^N-L^ (*p* < 0.001) group (Figure 4D–F).

Taking our results together, we proposed a model to explain that the pathophysiology of critical COVID-19 can be mediated by different pathways, depending on the TNF/IFN-γ levels (Figure 5). Thus, the viral nucleic ssRNA activates TLR7 and/or TLR8 signaling, leading to one of two scenarios. The first, excessive activation of MYDD88, IRF7, and NFκB, results in massive production of proinflammatory cytokines (including TNF and IFN-γ), enhancing the cytokine storm. Moreover, tmTNFR1 can interact with TNF (by an autocrine or paracrine effect), and cell death is activated, confirmed by the high levels of FADD and TRADD and the high levels of cytotoxic molecules. Accordingly, patients with this profile have cells in hyperactive status, which could be responsible for causing critical illness (Figure 5, left). In the second scenario, basal activation occurs where the presence of MYDD88, IRF7, and NFκB is residual; there is low cytokine production (including TNF and IFN-γ); and the presence of cytotoxic molecules is not increased. Accordingly, these patients with low-to-normal levels of TNF/IFN-γ have cells in hypoactive status, which could also contribute to inducing critical illness (Figure 5, right). This status opens a new question that should be considered in subsequent studies about whether this hypoactivation affects the presence of IFN-stimulated genes and what could be responsible for the critically ill in this context.

## 3. Discussion

We evaluated a functional profile based on the cytokine production and systemic soluble molecules of COVID-19 patients classified based on TNF/IFN-γ levels. The TNF^H^IFNγ^H^ group had immune cells in hyperactive status that is characterized by high production of cytotoxic molecules, proinflammatory cytokines, and TNFR1, suggesting that this microenvironment could induce PANoptosis and, consequently, poor outcomes in the patients, even death. In contrast, the TNF^N-L^IFNγ^N-L^ group displayed a low ability to produce proinflammatory cytokines and cytotoxic molecules, and molecules involved in the TLR7/TLR8 signaling were affected.

Our findings show that COVID-19 patients undergoing IMV are mainly associated with the group with high systemic TNF levels; however, only the group with increased TNF/IFN-γ levels had significantly increased molecules associated with signaling to produce proinflammatory cytokines, which may enhance the development of the cytokine storm and cell death.

We demonstrated that TNF/IFN-γ levels are the leading players in determining the activation of a specific pathway that favors massive inflammation to induce critical illness, as observed in the TNF^H^IFNγ^H^ group. In contrast, the TNF^N-L^IFNγ^N-L^ group exhibited decreased proinflammatory production, suggesting that the severity in this group has a different physiopathology.

Studies have established that increased levels of IL-1β, IL-2, IL-6, IL-10, IFN-γ, TNF, IFN-γ-inducible protein 10 (IP-10), granulocyte macrophage-colony stimulating factor (GM-CSF), and monocyte chemoattractant protein-1 (MCP-1) correlate with COVID-19 severity [16,17]. Additionally, high levels of soluble TNFR1 are associated with mortality [13,18]. In this way, TNF/TNFR1 signaling is crucial in mediating inflammation and cell death; binding of TNF to TNFR1 promotes and exacerbates apoptosis, necroptosis, and pyroptosis [19]. These three programmed cell death (PCD) types are activated in parallel through an assembled protein complex, the PANoptosome, to induce PCD PANoptosis [20,21].

Our findings suggest that TNF interacts with TNFR1, contributing to cell death. Immune cells from COVID-19 IMV patients (in which TNF^H^IFNγ^H^ and TNF^H^IFNγ^N-L^ are included), but not NIMV patients, express the transmembrane form of TNFR1. IMV was mainly required in COVID-19 patients with high TNF levels (TNF^H^IFNγ^H^ and TNF^H^IFNγ^N-L^). These results suggest that IMV patients with high TNF and IFN-γ levels (like TNF^H^IFNγ^H^) have increased frequency of immune cells positive for TNFR1 and display a profile with high circulating levels of cytokines and cytotoxic molecules, suggesting an exacerbated immune profile.

The host genetic background is a key factor in human susceptibility to or resilience against viral infections and disease [10]; the relationship between genetic variations and the immune response against SARS-CoV-2 has been investigated. SNPs in human leukocyte antigen (HLA), TLRs, and ACE2, among other genes, are critical factors considered as a risk for contributing to COVID-19 susceptibility and severity [22,23,24]. Some reports have revealed an association of rare *TLR7* gene variants with COVID-19, specifically in young male patients who required supplemental oxygen, suggesting the involvement of these TLR7 variants in the severity of the disease [25,26,27]. Despite the small sample size, our study identified the C allele of the rs3853839 SNP in *TLR7* as dominant in the TNF^H^IFNγ^H^ group; in this sense, recently, it has been proposed that rs3853839 presence increases the risk of severe outcomes of COVID-19 [28]. While haplotypes GGG and ACA were found for *TLR8*, with a dominance of haplotype ACA only in the TNF^H^IFNγ^N-L^ group, these results suggest that these genotypes may be associated with high TNF/IFN-γ levels during COVID-19. Also, we observed that allele T of rs2285666 in *ACE2* is present mainly in patients with the highest TNF levels. In contrast, previous reports have indicated that the G-allele in *ACE2* is related to susceptibility to and the severity of COVID-19, highlighting a controversial effect on severity [29,30].

Of note, Made et al. found that loss of function in *TLR7* variants is a determinant of and associated with severity and death in COVID-19 patients [27]; furthermore, SNPs in *TLR7* and *TLR8* have been related to COVID-19 in females (A-allele of the TLR7 rs179009 SNP) and to disease severity and in males (rs3764880) to comorbid diseases [31]. *TLR8* gene variants have been associated with the severity of some viral infections or, conversely, with a protective effect modulating the cytokine response (rs3764880) in HIV but with disease progression in patients with tuberculosis [32,33,34]. In contrast, some other *TLR8* variants are unrelated to COVID-19 severity [35].

All these contradictory findings show that the vast genetic diversity in different populations is unique, and it impacts the TLR7-/TLR8-mediated response against COVID-19 infection. TLR8 variants and their agonists have been involved in cell activation, increasing IFN-γ production in viral infections [36,37]. In our study, *TLR7* and *TLR8* variants were more closely related to IFN-γ levels. Patients with the highest TNF/IFN-γ levels (TNF^H^IFNγ^H^) had higher levels of molecules associated with TLR7 and TLR8 signaling (MYD88, NFκB, and IRF7) and cell death mediated by the TNF/TNFR1 pathway (TRAF2, FADD, and TRADD) than patients with the lowest TNF/IFN-γ levels.

Results showed that systemic TNF and IFN-γ levels in patients are helpful in dividing patients into those who have an excessive response to cytokine and cytotoxic molecule production (TNF^H^IFNγ^H^) and those who have less ability to activate said response (TNF^N-L^IFNγ^N-L^), suggesting that the pathophysiology of critical COVID-19 can be mediated in different pathways.

## 4. Materials and Methods

### 4.1. Ethics Approval and Consent to Participate

This protocol was approved by the Ethical Researcher and Investigation Committees of the Instituto Nacional de Enfermedades Respiratorias Ismael Cosio Villegas (INER; protocol numbers C41-20, C53-20, and B09-22) and the Hospital Central Militar (HCM; protocol number C.Inv.039). All individuals signed a consent letter to participate in this study. All procedures agreed with the 1964 Helsinki Declaration and the ethical standards of the Institutional Ethics Committees, and samples were processed according to Institutional Biosafety Guidelines.

### 4.2. Study Population

A total of 150 COVID-19 patients were enrolled from the Instituto Nacional de Enfermedades Respiratorias (INER) and the Hospital Central Militar (HCM) in Mexico City during the first pandemic wave, from May to September 2020. Reverse transcription polymerase chain reaction (RT-PCR) of nasopharyngeal swabs confirmed SARS-CoV-2 infection. Nine healthy individuals who tested negative for SARS-CoV-2 by RT-PCR were used as controls. Both controls and COVID-19 patients were naive to the COVID-19 vaccines.

To classify patient groups, we used a COVID-19 severity scale that we had previously reported, which aligns with the WHO definitions [13,15,38,39]. Briefly, the classification process was based on institutional clinical practices and the literature, including the institutional Picture Archiving and Communication System (PACS), the CARE score, the National Institutes of Health COVID-19 Treatment Guidelines, and the Radiographic Assessment of Lung Edema (RALE) score. Chest radiographs and tomography scans were reviewed using PACS, and the classification of disease severity as mild, moderate, or critical was conducted by three pulmonologists and one thoracic radiologist, who reviewed all patient data. Severity was defined as follows: mild (1–2 points), moderate (3–6 points), and critical (>6 points). For this study, the severity scale was assigned by the institutional clinical staff, who also considered oxygen saturation, the respiratory rate, and relevant biochemical parameters.

According to international standards, the institutional clinical staff recommended invasive mechanical ventilation (IMV) for patients presenting with dyspnea, a respiratory rate of ≥30 breaths per minute, blood oxygen saturation ≤90%, and a ratio of arterial oxygen partial pressure (PaO_2_, in mmHg) to fractional inspired oxygen (FiO_2_, expressed as a fraction) (PaO_2_/FiO_2_) ≤ 300. Patients who did not meet these criteria were managed without IMV (NIMV).

As shown in Appendix A, 16 patients with IMV, 9 with NIMV, and 9 HD were considered for cell phenotype evaluation by flow cytometry and systemic cytotoxic molecules; 125 patients were included to analyze cytokines, SNPs, and molecules associated with TLR and TNF/TNFR signaling at the transcriptional level.

The 25 COVID-19 patients were matched with demographic characteristics similar to the 125 patients (Appendix A). The IMV group (n = 16) had 50% males, with a median age of 53 years (IQR 44–59) and a median leukocyte count of 11.9 (IQR 8.82–14.33), and 69% displayed leukocytosis. In the NIMV group (n = 9), 88% were males, with a median age of 48 years (IQR 36–75) and a median leukocyte count of 9.8 (IQR 7.9–13.4), and 22% presented with leukocytosis. Of note, D-dimer and LDH values were higher in the IMV than in the NIMV group.

### 4.3. Sample Preparation

Blood samples were obtained upon individuals’ arrival at the hospital and before initiating therapy; plasma peripheral blood mononuclear cells (PBMCs) were recovered using BD Vacutainer tubes (BD Biosciences, Franklin Lakes, NJ, USA). PBMCs were isolated by the standard LymphoprepTM (Accurate Chemical-Scientific, New York, NY, USA) centrifugation gradient within one hour of the blood draw and were subsequently cryopreserved. Plasma was obtained and stored at −70 °C until use. Additionally, blood samples were stored at −70 °C in DNA/RNA Shield solution (Zymo Research, Irvine, CA, USA) to the posterior obtention of RNA and DNA. Briefly, samples were diluted in a proportion of 1 to 3 volumes of solution, mixed, and then incubated at room temperature for cell lysis and to inactivate the coronavirus; the homogenate was frozen to preserve the stability of nucleic acids.

### 4.4. Quantification of TNF and IFN-γ Soluble Levels

In 125 of the 150 COVID-19 patients, plasma TNF and IFN-γ levels were quantified by enzyme-linked immunosorbent assay (ELISA) following the manufacturer’s protocols with the Human TNF-Alpha Duoset Elisa Kit (R&D Systems; Minneapolis, MN, USA) and ELISA MAX™ Deluxe Set Human IFN-γ (BioLegend, San Diego, CA, USA). A total of 100 μL of each plasma sample was used for each cytokine evaluation per duplicated sample, a tetramethylbenzidine colorimetric substrate was used to develop the blue color, and optical density (450 nm) was measured using a microplate reader (Imark, Bio-Rad, Hercules, CA, USA). The two cytokines were quantified by comparison with their corresponding standard curves using a 4-parameter logistics curve-fitting algorithm (sigmoidal 4PL) and a coefficient of determination (R^2^) > 0.90.

### 4.5. Classification of COVID-19 Patients According to TNF and IFN-γ Levels

Similar to our previous study, we used TNF and IFN-γ levels of the control group (n = 9) for classifying COVID-19 patients [15]. High (H) or normal–low (N-L) levels were determined by the mean TNF and IFN-γ levels of the control group (n = 9), which was 10.2 pg/mL for TNF and 12.6 pg/mL for IFN-γ.

TNF^H^ and TNF^N-L^ were defined as TNF > 10.2 pg/mL and TNF ≤ 10.2 pg/mL, respectively. IFN-γ^H^ and IFN-γ^N-L^ were defined as IFN-γ > 12.6 pg/mL and IFN-γ ≤ 12.6 pg/mL, respectively. Of the 150 COVID-19 patients, 125 were classified as follows: TNF^H^IFN-γ^H^ (n = 22), TNF^H^IFN-γ^N-L^ (n = 20), TNF^N-L^IFN-γ^H^ (n = 44), and TNF^N-L^IFN-γ^N-L^ (n = 39). The remaining 25 were divided into IMV (n = 16) and non-IMV (n = 9) groups according to the use of IMV (Appendix A).

### 4.6. Cytotoxic Molecule Quantification by Multiplex Bead-Based Assay

IL-2, IL-4, IL-10, IL-6, IL-17A, TNF, sFas, sFasL, IFN-γ, granzyme A, granzyme B, perforin, and granulysin were quantified in the plasma using the LEGENDplex^TM^ Human CD8/NK panel according to the manufacturer’s instruction (BioLegend, CA, USA). In total, 10,000 events were acquired from bead A and B gates via the FACS Aria II flow cytometer (BD Biosciences) and analyzed by LEGENDplex™ software version 2024-06-15 2019 Qognit (https://legendplex.qognit.com, access date 10 July 2024).

### 4.7. Measurement of TNFR1 and TNFR2 Expression by Flow Cytometry

The transmembrane expression of TNFR1 and TNFR2 was evaluated on the cell surface of CD4+ and CD8+ T cells and monocytes (CD14+ cells) for IMV and NIMV groups. PBMCs were prepared to evaluate cell surface markers’ expression using monoclonal antibodies (mAbs) to CD2 (APCH7/APCCY7), CD3 (Brilliant Blue 700), CD4 (Brilliant Violet 510), CD8 (Brilliant Ultraviolet 563), CD14 (Brilliant Violet 510), TNFR1 (APC), and TNFR2 (PE/Dazzle 594). mAbs were provided by BioLegend (San Diego, CA, USA) and BD Bioscience (San Jose, CA, USA); details for mAbs are shown in Appendix A. The cells used for the Fluorescence Minus One (FMO) condition and individual stains were stained and acquired in parallel to identify background levels of staining; dead cells were omitted using the viability staining Zombie Red Dye solution (BioLegend, San Diego, CA, USA).

Data were acquired using a FACS Symphony flow cytometer (BD Biosciences) equipped with FACSDiva 6.1.3 software (BD Biosciences, San Jose, CA, USA). At least 100,000 events were acquired per sample and condition following the recommended protocols [40,41]. The flow cytometry data file (FCS) was analyzed using FlowJo™ v10.6.1 (Flow Jo, LLC, Ashland, OR, USA). The analysis strategy involved gating singlet cells based on forward scatter (FSC-A versus FSC-H) and selecting viable cells using Zombie Red-negative staining. Subsequently, the CD2+CD3+ gate was applied to identify the lymphocyte population. The expression of CD4 and CD8 was then determined to define the immunophenotypes for CD2+CD3+CD4+ and CD2+CD3+CD8+ T cells, followed by the evaluation of TNFR1 and TNFR2 expression. For monocytes, the gating strategy began by selecting viable cells, followed by identifying CD2- and CD3- cells. Within this population, CD14+ cells were gated, and the expression of TNFR1 and TNFR2 on CD14+ monocytes was analyzed (Appendix A).

### 4.8. Genotyping and Analysis of Gene Expression by Quantitative Real-Time PCR

A total of 125 COVID-19 patients were tested for SNPs in TLR7, TLR8, and ACE2 to identify whether the frequency increased in either COVID-19 group, stratified based on the patients’ TNF and IFN-γ levels; moreover, the transcriptional levels of *MYD88*, *NFKB1*, *IRF7*, *TRAF2*, *FADD*, and *TRADD* were assessed to clarify the status of molecules involved in TLR signaling or related to the TNF/TNFR1 axis.

Genomic DNA and total RNA were isolated from PBMCs preserved at −70 °C in DNA/RNA Shield solution (Zymo Research, Irvine, CA, USA) using the Quick-DNA^TM^ Miniprep Kit (Zymo Research, Irvine, CA, USA) for DNA and the RNeasy Micro Kit (Qiagen, Hilden, Germany) for RNA according to the manufacturer’s instructions. Before genotyping or cDNA synthesis, purity, quality, and concentration were analyzed with a Qubit 2.0 Fluorometer (Life Technologies, Waltham, MA, USA).

SNPs were detected using seven TaqMan SNPs Genotyping assays specific to angiotensin-converting enzyme 2 (ACE2), *TLR7*, and *TLR8* located in the X chromosome—one in *ACE2* (C_188785287_10; rs2285666, splice donor region), three in *TLR7* (C_2259574_10, rs179008, missense variant; C_2259575_10, rs179009, intron variant; and C_2259573_10, rs3853839, 3′UTR), and three in *TLR8* (C_27497635_10, rs3761624, intron variant; C_2183829_10, rs3764879, intron variant; and C_2183830_10, rs3764880, start lost)—in a reaction volume of 10 μL following the manufacturer’s instructions.

Genomic DNA was removed using the RNA-Free DNAse Set (Qiagen, Hilden, Germany) in the RNA samples for gene expression analysis. Following the manufacturer’s guidelines, 270 ng of total RNA was used for first-strand cDNA synthesis using the High Capacity cDNA Reverse Transcription Kit (Applied Biosystems, Waltham, MA, USA). Gene expression was assessed using TaqMan probes for the target genes, Fas-associated via death domain (*FADD*, Hs00538709_m1), Tradd-TNFRSF1A associated via death domain (*TRADD*, Hs00601065_g1), Traf2-TNF receptor associated factor 2 (*TRAF2*, Hs00184192_m1), Myd88-myeloid differentiation primary response 88 (MyD88, Hs01573837_g1), Nfkb1-nuclear factor kappa B subunit 1 (*NFKB1*, Hs00765730_m1), and Irf7-interferon regulatory factor 7 (IRF7, Hs01014809_g1). The 18S ribosomal RNA gene (18S, Hs03928990_g1) and β-actin (*ACTB*, Hs01060665_g1) were used as endogenous controls. Single reactions were prepared with the Maxima Probe/ROX qPCR Master Mix (Thermo Fisher Scientific, Waltham, USA), cDNA was diluted fourfold (18 ng/μL to 4.5 ng/μL), and amplifications were performed per duplicate under the following thermal conditions: 95 °C for 10 min, followed by 40 cycles of 60 °C for 1 min and 95 °C for 15 s. Relative gene expression was determined using the ΔΔCT method to calculate the n-fold change for each target gene in COVID-19 patients’ groups. Results were normalized with ACTB and 18S, relative to a control group of nine healthy donors (RQ = 1).

### 4.9. Statistical Methods

The Shapiro–Wilk test was used to evaluate the normality of the distribution of numerical variables. The Mann–Whitney U test was used to compare two groups, and multiple comparisons were performed with the Kruskal–Wallis test and corrected using Dunn’s test. The chi-square test was performed for polymorphism variables. A *p*-value of <0.05 was considered statistically significant for all tests.

Since *TLR7* and *TLR8* genes are in the X chromosome, the Hardy–Weinberg equilibrium (HWE) was assessed for all polymorphisms only in the female gender.

All analysis was performed using GraphPad 10.4.1 (GraphPad Software, Inc., San Diego, CA, USA) and STATA v.16 (StataCorp LLC, College Station, TX, USA).

## 5. Conclusions

Plasmatic TNF/IFN-γ levels in COVID-19 patients could indicate two pathological processes leading to a critical illness. Increased levels of both cytokines are associated with uncontrolled activation of the TLR7/TLR8 pathway, which favors a cytokine storm and cell death. In contrast, normal or low levels of these cytokines indicate a lack of TLR7-/TLR8-dependent cellular response associated with a hypoactive cell status, where the mechanisms for the critically ill are unclear. These findings provide evidence that could be helpful in the establishment of new therapeutic strategies based on TLR7 and TLR8 signaling modulation.

## Figures and Tables

**Figure 1 ijms-26-01139-f001:**
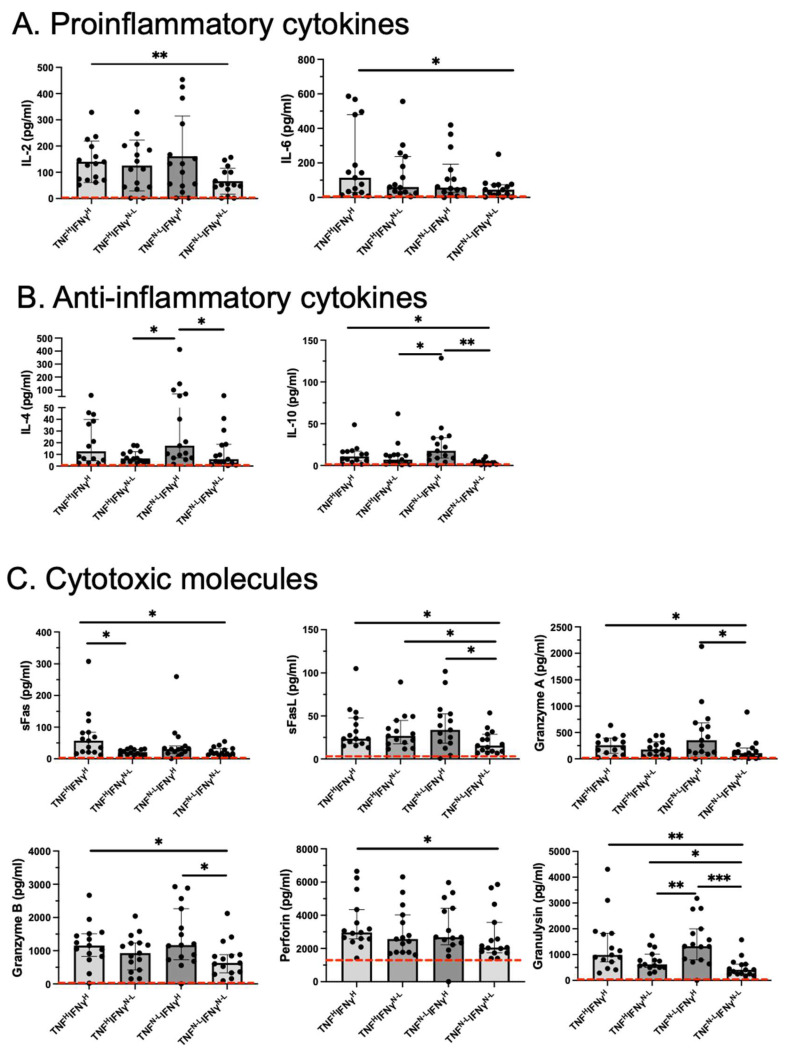
The ability of TNF^N-L^IFNγ^N-L^ to produce cytokines and cytotoxic molecules was affected compared to TNF^H^IFNγ^H^. Cytokines and cytotoxic molecules of COVID-19 patients based on TNF and IFN-γ levels (n = 125). The dotted red lines indicate the median levels of the variables in controls (n = 9). Soluble levels of (**A**) proinflammatories (IL-2 and IL-6), (**B**) anti-inflammatory (IL-4 and IL-10) cytokines, and (**C**) cytotoxic molecules (sFas, sFasL, granzyme A, granzyme B, perforin, and granulysin). Data are represented as median and IQR (25–75) values, and each dot represents a patient. The Kruskal–Wallis test was performed for statistical comparisons; * *p* < 0.05, ** *p* < 0.01, *** *p* < 0.001.

**Figure 2 ijms-26-01139-f002:**
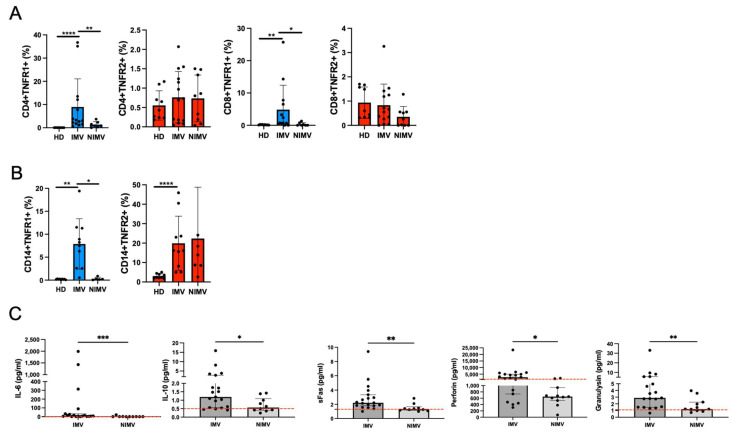
Patients with IMV have increased immune cells positive for TNFR1 and have systemic high levels of cytokines and cytotoxic molecules. Mononuclear cells (for flow cytometry) and plasma (for LEGENDplex^TM^) from 16 COVID-19 patients who received IMV, 9 who did not (NIMV), and 9 healthy donors (HDs). (**A**) Frequency of CD4+ or CD8+ co-expressing TNFRs. (**B**) CD14 co-expressing TNFRs. (**C**) Cytokines and cytotoxic molecules. Dotted red lines indicate the median levels of items in controls (n = 9). Data are represented as median and IQR (25–75) values, and each dot represents a patient. The Kruskal–Wallis test was performed for statistical comparisons; * *p* < 0.05, ** *p* < 0.01, *** *p* < 0.001, **** *p* < 0.0001.

**Figure 3 ijms-26-01139-f003:**
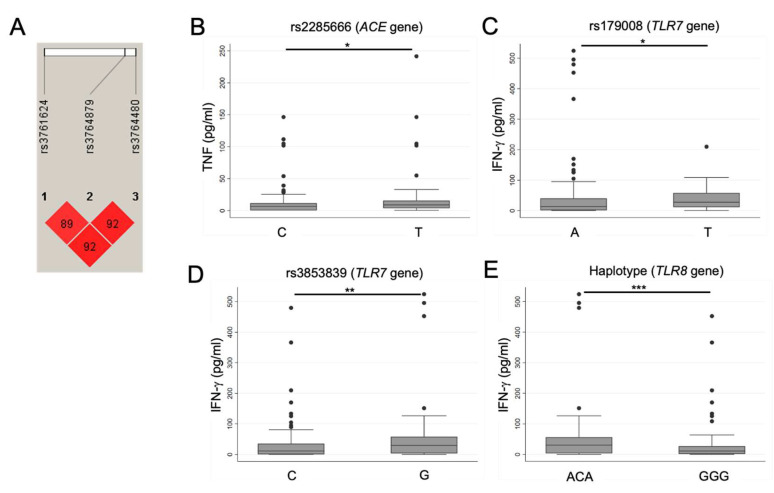
Association of SNPs with TNF or IFN-γ plasma levels. (**A**) Linkage disequilibrium of rs3761624, rs3764879, and rs3764480 of the *TLR8* gene. Levels of TNF (pg/mL) and IFN-γ (pg/mL). (**B**) The rs2285666 SNP of the *ACE2* gene. (**C**) The rs179008 SNP of the *TLR7* gene. (**D**) The rsrs3853839 SNP of the *TLR7* gene. (**E**) Haplotypes of the *TLR8* gene. The numbers inside the diamonds indicate D’ pairwise. Data are represented as median and IQR (25–75) values. The Kruskal–Wallis test was performed for statistical comparisons; * *p* < 0.05, ** *p* < 0.01, *** *p* < 0.001.

**Figure 4 ijms-26-01139-f004:**
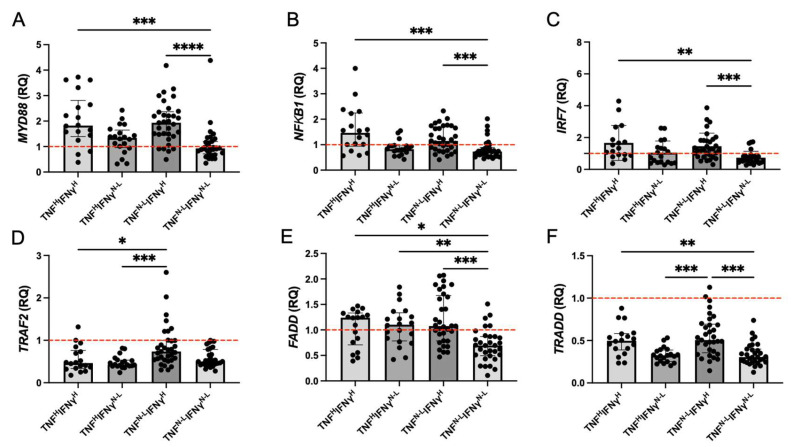
TNF^H^IFNγ^H^ patients have more molecules associated with TLR signaling and cell death mediated by the TNF/TNFR1 pathway than the TNF^N-L^IFNγ^N-L^ group. The relative quantification (RQ) of *MYD88* (**A**), *NFKB1* (**B**), *IRF7* (**C**), *TRAF2* (**D**), *FADD* (**E**), and *TRADD* (**F**) genes was measured at the transcriptional level. The RQ of the HD group was normalized to 1 (dotted red line). Data are represented as median and IQR (25–75) values, and each dot represents a patient. The Kruskal–Wallis test was performed for statistical comparisons; * *p* < 0.05, ** *p* < 0.01, *** *p* < 0.001, **** *p* < 0.0001.

**Figure 5 ijms-26-01139-f005:**
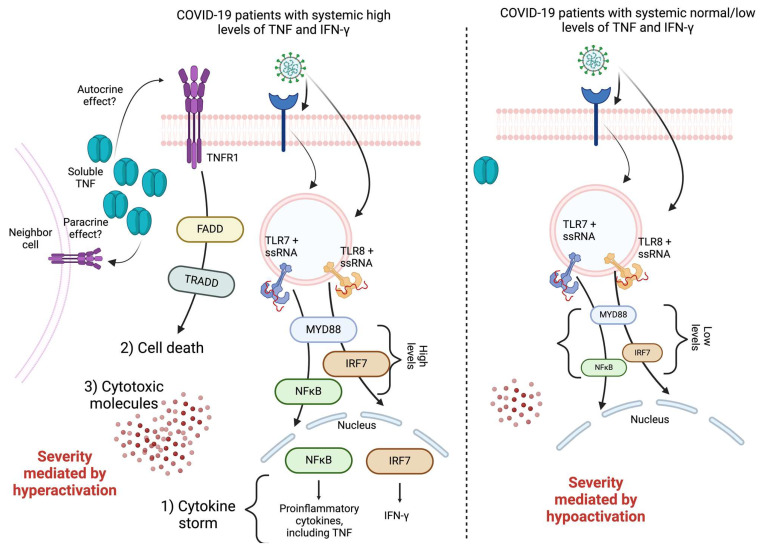
TLR7/TLR8 signaling triggers a proinflammatory response, a cytotoxic profile, and cell death favored by high TNF and IFN-γ levels in COVID-19 patients, causing severe disease mediated through hyperactivation. In contrast, disease severity is also caused by hypoactivation; COVID-19 patients display normal or low TNF and IFN-γ levels and lower expression of molecules related with TLR7/8 signaling and critical molecules involved in cell death. The figure was created in BioRender.

**Table 1 ijms-26-01139-t001:** Baseline characteristics of COVID-19 patients classified according to TNF and IFN-γ levels (n = 125).

Group	Total (n = 125)	TNF^H^IFNγ^H^ (n = 22)	TNF^H^IFNγ^N-L^ (n = 20)	TNF^N-L^IFNγ^H^ (n = 44)	TNF^N-L^IFNγ^N-L^ (n = 39)	*p **
Age (years)	55 (45–64)	56 (50–65)	58 (49–69)	57 (42–65)	53 (37–63)	0.4597
Sex						
Male, n (%)	86 (69)	17 (77)	16 (80)	27 (61)	26 (67)	0.9813
Leukocyte, ×10^9^/L	8.5 (6.3–11.48)	9.2 (6.57–13.45)	8.6 (6.4–13.13)	7.8 (5.14–9.47)	8.4 (6.95–11.20)	0.0778
D-dimer (ng/mL)	0.88 (0.4–2.4)	1.5 (0.2–2.2)	3.1 (1.2–6.2)	0.6 (0.3–2.3)	0.9 (0.4–2.3)	0.0042 **
LDH (U/L)	361 (292.3–545.8)	417.0 (288.3–676.5)	339.0 (232–365)	435.0 (304–676.5)	345.0 (269–534)	0.0732
Disease severity						<0.0001
Mild, n (%)		2 (9)	4 (20)	15 (34)	9 (23)	
Severe, n (%)		7 (32)	11 (55)	20 (45.5)	15 (38.5)	
Critical, n (%)		13 (59)	5 (25)	9 (20.5)	15 (38.5)	
IMV, n (%)		16 (73)	17 (85)	9 (20.5)	8 (20.5)	<0.0001

Data are presented as a number (percentage) or the median (interquartile range). LDH, lactate dehydrogenase. * *p*-Values were calculated using the Kruskal–Wallis or chi-square test among the four groups. ** The median D-dimer level in the TNF^H^IFNγ^N-L^ group was significantly higher than that in the TNF^N-L^IFNγ^H^ (*p =* 0.0018) and TNF^N/L^IFNγ^N-L^ (*p =* 0.0303) groups. IMV, patients who received invasive mechanical ventilation.

## Data Availability

The data generated and analyzed in this study are available from the corresponding author upon reasonable request.

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
