# Peer review of "TNF/IFN-γ Co-Signaling Induces Differential Cellular Activation in COVID-19 Patients: Implications for Patient Outcomes"

_ijms, 2025, doi:10.3390/ijms26031139_

Round 1
Reviewer 1 Report
Comments and Suggestions for Authors
This is not a well written paper. There is a lot of unnecessary explanations in the results, which corresponds to the method and discussion. The description of the method is not clear.
In my opinion, figure 6 seems to represent the purpose of this study. Therefore, I recommend moving Figure 6 to the result section. Revise the main draft and title according to the modified purpose and resubmit it.

Comments on the Quality of English LanguageExtensive English editing is required.
Author Response
This is not a well written paper. There is a lot of unnecessary explanations in the results, which corresponds to the method and discussion. The description of the method is not clear.
Dear Reviewer, We appreciate your comments. Below, we reply to your observations point by point, and we followed your suggestions throughout the manuscript.
In my opinion, figure 6 seems to represent the purpose of this study. Therefore, I recommend moving Figure 6 to the result section. Revise the main draft and title according to the modified purpose and resubmit it.
Comments on the Quality of English Language
Extensive English editing is required.
The new version was modified based on the comments in the PDF file, and an English expert revised the language.
Abstract: Line 24 was rewritten. Lines 25-26, The mean of groups was simplified to be easily read (following your recommendation in the method section for patient classification). We appeal to your comprehension to maintain the description of “Gp” in this section because it is easier to understand the group’s description using molecules/levels.
Introduction: line 39, “PRR” eliminated. In lines 50-51, changes are done. Lines 52-54 and 66-67 were moved from the discussion per your suggestion. Lines 76-80 were rewritten.
Results: 2.1 section, description groups were moved to methods, IQRs were added, and the demographic data table was included. 2.2 section, lines 109-113 were rewritten following your comments (in the previous version, they were lines 112-120). Previous Figure 2 is the current supplementary Figure S2, described on lines 127-130. In the 2.3 section, according to your comments, some phrases were removed, and baseline characteristics of IMV and NIMV groups were included as supplementary Table S1. In the 2.4 section, the indicated phrase was removed; the n=122 for allelic frequencies is correct (indicated in the previous line 178) as it is indicated in the footnote of Table S2: “Three of the 125 subjects assigned to genotyping analysis were excluded because of the poor DNA concentration; thus, 122 were genotyped, 84 men and 38 women”. To avoid confusion, this note was included in the supplementary Figure S1, and the requested n was included in the main manuscript (lines 180 and 184, respectively). In the 2.5 section, the indicated phrase was removed, and IQRs were included. In the 2.6 section, the indicated phrase was removed, and Figure 6 (new Figure 5) was included in this section and removed from the discussion.
Discussion: All indicated phases were deleted in this section, and relocalized where suggested.
Methods: Indicated paraphrases from results were moved to this section. In the 4.2 section, we provided more details about the patient classification, and the use of IMV according to the OMS standards; as was indicated, we used this classification in several reports (references 15, 17, 41, and 42, and additional reference https://doi.org/10.1186/s12890-023-02807-8 show also the use of these parameters to determinate IMV), it is on lines 329-346; regarding the number of patients that used IMV in our cohort of 125 patients, it is indicated the current supplementary Figure S2. In the 4.4 section, details of antibodies were included in the new supplementary Table S4; more details of the analysis strategy and reference for “100,000 events” were added (lines 384-391). 4.5-4.9 sections, your suggestion was done following your instructions.
Reviewer 2 Report
Comments and Suggestions for Authors
Dear Authors,
I Would like to thank you for your efforts for your manuscript entitled "The TNF/IFN- Co-signaling Induces a Differential Cellular Activation in COVID-19 Patients: Implications for Patient outcomes".
I have some comments that I wish it may help:
1- Healthy donors are 9; which is not even closer to the patient sample size. This makes your comparison weaker.
2- The flow cytometry panel is wrong since you have 2 markers with the same color. CD4 and CD14 with Brilliant violet 510. If you use them with 2 different tubes, this makes your analysis not strong enough because relative cell count comparisons is important.
3- Flow cytometry analysis scheme should be written in detail for reproducibility.
4- IMV and NIMV abbreviations are not clear.
5- Results of figure 1 is not correct. In GP1 the outliers of TNF and IFN-g change the mean value and standard deviation, and of course will change the analysis. You can take these samples with outliers and group them to be analyzed separtely.
6- Figure 4: SNP of ACE gene is not related to the disease rather than COVID-19 infection. There are many publications discussing that. How to justify that SNP of rs2285666 is related to TNF, please explain in details from literature.
7- Results section is so confusing and needs to be rewritten. I can advise to explain the results of each groups alone and mention only the important significant comparisons. In discussion, do not discuss results but explain your hypothesis in a story way and supported by your data.
I encourage the authors to do the corrections to emerge a piece of art.
Good luck.
Author Response
I Would like to thank you for your efforts for your manuscript entitled "The TNF/IFN-g Co-signaling Induces a Differential Cellular Activation in COVID-19 Patients: Implications for Patient outcomes".
Dear Reviewer, we appreciate all your comments, which have improved our study. Below, we reply to your observation point by point.
1- Healthy donors are 9; which is not even closer to the patient sample size. This makes your comparison weaker.
Dear Reviewer, you are correct regarding the small number of HDs. However, this group was used only as a basic reference to divide our COVID-19 groups. As indicated in reference 17, we used a separate HD cohort to determine "normal" levels, which aligns with the findings in this study, including flow cytometry results, because the more important conclusion is that IMV has different levels than NIMV. Thus, meaningful comparisons were made between COVID-19 groups to develop a model explaining the pathophysiology of critical illness. Thus, we believe the small sample size of HDs does not affect our main conclusion that severity in Gp1 and Gp4 is mediated by different pathways. Consequently, the small sample size in HD is not a limitation of the study.
2- The flow cytometry panel is wrong since you have 2 markers with the same color. CD4 and CD14 with Brilliant violet 510. If you use them with 2 different tubes, this makes your analysis not strong enough because relative cell count comparisons is important.
Dear Reviewer, your observation regarding shared fluorochromes is correct. However, analyzing different cell subsets in separate tubes is standard and does not weaken the analysis. In one tube, we evaluated TNFR1 and TNFR2 expression on T cells, and in another, on CD14+ monocytes, ensuring no T cell contamination by gating CD2-CD3- cells. Our aim was not to compare lymphocyte and monocyte counts but rather to evaluate marker expression. This clarification has been added to the Methods section (lines 382–387) and detailed in Table S4.
3- Flow cytometry analysis scheme should be written in detail for reproducibility.
Thank you for this suggestion. The revised Methods section now includes detailed descriptions of the analysis strategy.
4- IMV and NIMV abbreviations are not clear.
We apologize for the lack of clarity. The revised manuscript now clearly defines the abbreviations for IMV and NIMV.
5- Results of figure 1 is not correct. In GP1 the outliers of TNF and IFN-g change the mean value and standard deviation, and of course will change the analysis. You can take these samples with outliers and group them to be analyzed separtely.
Dear Reviewer, the indicated data points are not statistical outliers. Statistical tests confirmed their inclusion. Moreover, even if they were outliers, their presence did not affect our group classification, as patients were categorized individually based on cytokine levels compared to HD levels (red dotted line). These levels are consistent with our previous report (reference 17). The group definitions have been clarified in section 4.6 of the Methods.
6- Figure 4: SNP of ACE gene is not related to the disease rather than COVID-19 infection. There are many publications discussing that. How to justify that SNP of rs2285666 is related to TNF, please explain in details from literature.
Diverse evidence demonstrates that the ACE2 variant rs2285666 modulates inflammatory responses mediated by TNF. Haga et al. (PNAS, doi:10.1073/pnas.0711241105) showed TNF-α formation in response to SARS-CoV spike protein-induced ACE2 shedding. Gawish et al. (eLife, doi:10.7554/eLife.74623) reported that genetic ACE2 deficiency prevented mouse COVID-19 and demonstrated cytokine-driven disease severity dependent on ACE2. Additionally, studies suggest ACE2 mediates inflammatory cytokines through NF-κB signaling induced by SARS-CoV-2 spike protein interaction (doi:10.1128/CMR.00299-20; doi:10.3390/biomedicines10020242; doi:10.3345/cep.2024.00941). Further studies are needed to confirm this pathway.
7- Results section is so confusing and needs to be rewritten. I can advise to explain the results of each groups alone and mention only the important significant comparisons. In discussion, do not discuss results but explain your hypothesis in a story way and supported by your data.
Thank you for your suggestion. The Results section has been rewritten for clarity. Diverse sections of the discussion were removed, and we consider the new version focused on hypothesis building, supported by our data.
I encourage the authors to do the corrections to emerge a piece of art.
We appreciate your valuable feedback. Several changes were made to the manuscript following your suggestions, enhancing its clarity and rigor.
Round 2
Reviewer 1 Report
Comments and Suggestions for Authors
There is still a lot of methodological description in the results, and the tables and figures need to be revised to be more concise. The English description is too verbose. Be straightforward to understand.

Comments on the Quality of English LanguageThe English could be improved to more clearly express the research.
Author Response
There is still a lot of methodological description in the results, and the tables and figures need to be revised to be more concise. The English description is too verbose. Be straightforward to understand.
peer-review-43478856.v1.zipThis is not a well written paper. There is a lot of unnecessary explanations in the results, which corresponds to the method and discussion. The description of the method is not clear.
Dear Reviewer, We appreciate your comments. Below, we reply to your observations point by point.
Followed point by point the suggestion in the PDF file (delete or rewrite):
Abstract: Lines 25-26, following your recommendation groups, were written in the full description; we consider that using a complete description is not easy to read, but as you suggested, it is modified in the abstract and throughout the manuscript.
Introduction: Indicated phrases were removed or rewritten.
Results: 2.1 section, Figure 1A was removed, the first paragraph was rewritten, and data of supplementary 1 was added to Table 1. All suggested changes for Table 1 were made, and the same changes were considered for Table S1. Males and females were included in the analysis for sex to calculate the p-value using the chi-squared test, which was the same case for the analysis of IMV versus NIMV. 2.2 section, the data description followed the reviewer’s suggestion. Consequently, the comparison between other groups is not described in the document (for instance, group 2 versus 3, even if there are statistical differences); the description is limited to comparing Gp 1 and Gp4; Figure 1 was changed according to suggestion. In the 2.3 section, the new headline indicates only the result with 25 patients; data from supplementary 1 was added to Table 1; respectfully, regarding previous lines 131-133, we consider it essential to indicate that those results are from a different cohort, it was summary and maintained. Figure 2 was adjusted following the reviewer’ suggestion. In the 2.6 section, “Gp” is changed by the complete description.
Discussion: “Gp” is changed by the complete description.
Methods: In the 4.4 section, supplementary figure S2 is included. The 4.4 section is the new 4.7, 4.5 is the new 4.4, 4.6 is the new 4.5.
Reviewer 2 Report
Comments and Suggestions for Authors
Dear Authors,
I would like to thank the authors for the corrections are performed to the manuscript.
Although many points have been covered correctly, but there are some points are still not accurate for example:
1- outliers of data (figure1) either to be separated in a group or removed from the study. These samples may change the mean and standard deviation of results. The scattering may be disolved in large sample size.
2- immune cell analysis needs to be corrected, for example: monocytes are CD14+ which is correct, but there are monocytes who are CD14- which is called non-classical monocytes.
Usually, we make sequenential gating to discover monocytes as follows:
CD45+ CD3- CD19- CD33+ CD11b+ HLA-DR+ then go for monocytes markers which are CD14 and CD16.
I encourage authors to make more improvements to show the novelty of work rather than publishing incomplete and strong data.
Thank you.
Author Response
I would like to thank the authors for the corrections are performed to the manuscript.
Although many points have been covered correctly, but there are some points are still not accurate for example:
Dear Reviewer, We appreciate your comments. Below, we reply to your observations point by point.
1- outliers of data (figure1) either to be separated in a group or removed from the study. These samples may change the mean and standard deviation of results. The scattering may be disolved in large sample size.
R. All points observed as higher are not true outliers, and the statistical analysis indicated that it represents a normal variation of biological samples. However, this figure has been removed at the suggestion of Reviewer 1.
2- immune cell analysis needs to be corrected, for example: monocytes are CD14+ which is correct, but there are monocytes who are CD14- which is called non-classical monocytes.
Usually, we make sequenential gating to discover monocytes as follows:
CD45+ CD3- CD19- CD33+ CD11b+ HLA-DR+ then go for monocytes markers which are CD14 and CD16.
I encourage authors to make more improvements to show the novelty of work rather than publishing incomplete and strong data.
R. Yes, you are right; there are monocytes expressing different intensities of CD14, and a minority is CD14-. Using flow cytometry allows for a complete analysis of cell phenotypes. In this study, we did not consider the expression of other markers because our aim was limited to identifying TNFR1 and TNFR2 on CD14+ cells (as indicated in the graphic and section 2.3, we call them CD14+ monocytes) and on T cells (CD4+ and CD8+). We did not evaluate the expression on monocyte subsets or T cell subsets. Our scheme is straightforward for identifying CD4 and CD8, but we did not identify further subsets. We appreciate your understanding regarding the use of “Live CD2- CD3- CD14+” as a phenotype for general monocytes. When evaluating monocyte subsets, we used other markers such as HLA-DR, CD16, and CD11b (doi: 10.3390/ijms252011063; 10.1016/j.ijid.2023.11.032; 10.3389/fimmu.2022.1016472), but for general monocytes, we used the phenotype applied here (for example, doi: 10.1155/2021/6654220; 10.3390/ijms23010329; 10.1016/j.micpath.2021.104793; 10.1155/2015/984973).
Round 3
Reviewer 2 Report
Comments and Suggestions for Authors
Dear Authors,
I would like to thank for your corrections.
I wish you can take in consideration for the future correct immune analysis for more solid experimental design.
Good luck.